# Bcl-2 Family Members and the Mitochondrial Import Machineries: The Roads to Death

**DOI:** 10.3390/biom12020162

**Published:** 2022-01-19

**Authors:** Lisenn Lalier, François Vallette, Stéphen Manon

**Affiliations:** 1CRCINA, Inserm, Université de Nantes, Université d’Angers, 44000 Nantes, France; Francois.Vallette@univ-nantes.fr; 2LaBCT, Institut de Cancérologie de L’Ouest, 44800 Saint Herblain, France; 3UMR5095, CNRS, Université de Bordeaux, 33000 Bordeaux, France; manon@ibgc.cnrs.fr

**Keywords:** bcl-2 family, mitochondrial import machineries, apoptosis

## Abstract

The localization of Bcl-2 family members at the mitochondrial outer membrane (MOM) is a crucial step in the implementation of apoptosis. We review evidence showing the role of the components of the mitochondrial import machineries (translocase of the outer membrane (TOM) and the sorting and assembly machinery (SAM)) in the mitochondrial localization of Bcl-2 family members and how these machineries regulate the function of pro- and anti-apoptotic proteins in resting cells and in cells committed into apoptosis.

## 1. Introduction

Eukaryotes evolved from different symbiosis processes including a fundamental one thought to have occurred between alpha proteobacteria and archaeal superphylum [1]. Mitochondria have been central to endosymbiosis and have progressively developed, along more than a billion years of evolution, an interdependence to nuclei, another feature of eukaryotic cells. In particular, mitochondria have lost most of the DNA needed to code for all mitochondria-specific proteins and therefore need to import nuclei-encoded proteins. However, some membrane-embedded proteins are still translated by mitochondrial DNA [2]. Thus, the organelle exhibits both eukaryotic and prokaryotic features. Mitochondrial multi-proteins complexes are often considered as a eukaryotic invention [3]. Among these complexes, two can be considered as specific to eukaryotes—the translocases of the outer membrane (TOMs) and the members of the Bcl-2 family (Bcl-2s) complexes—respectively implicated in protein import of nuclei-encoded mitochondrial proteins and in the regulation of the mitochondrial apoptotic pathway [4]. Indeed, the control of cell death and survival by mitochondria could be considered as a tradeoff for the increasing dependence of the organelle during evolution toward nucleus-encoded instruction. Of note, even if mitochondrial protein import systems are well conserved among eukaryotes, the Bcl-2 family proteins are present only in metazoans [5]. Distant related Bcl-2s are present in lower eukaryotes, where mammalian members of this family can still be active. This suggests that Bcl-2s’ targets are well-conserved in most eukaryotes [6].

The canonical function of Bcl-2s is the regulation of the mitochondrial outer membrane permeabilization (MOMP), responsible for the cytosolic release of apoptogenic molecules and the subsequent activation of caspases, the final effectors of apoptosis. This implies the mitochondrial localization of these proteins, at least of some of them, yet several issues remain. Because of the utilization of different cellular models (tumoral or non-tumoral mammalian cells, heterologous models) of different systems of expression (endogenous vs. vector, tagged or not, native or carrying mutations) and of different experimental conditions (apoptotic or non-apoptotic, other stress), it is not exaggerated to write that every Bcl-2 family member has been found at least once in almost every subcellular compartment. These proteins are all encoded by the nuclear genome. Nevertheless, the structure of these proteins does not reveal how they could be imported to mitochondria. Additionally, it seems that the regulation of Bcl-2s’ localization participates to the regulation of their function and depends on the cellular context.

In this review, we first briefly describe the TOM and SAM families, implicated in the mitochondrial protein import and proper organization, and the Bcl-2 family members. We then focus on the structural common or specific characteristics of the Bcl-2s in order to question the need for a mitochondrial receptor and how TOMs and SAMs proteins have been involved in this function. We finally discuss the physiological significance of the interactions observed between Bcl-2s and their potential mitochondrial receptors in the functions of these proteins.

## 2. Protein Import across the Mitochondrial Outer Membrane

In bacteria, protein transport across membranes is supported by small β-barrel protein, and quite remarkably, these proteins can support both the import and export of very large proteins [7]. Mitochondrial protein import mechanisms in eukaryotes, from microorganisms to mammals, exhibit similar features and structures [8]. The outer membrane of mitochondria contains a highly specialized protein complex implicated in protein import into the organelle called TOM for translocase of outer membrane and a more specialized complex called sorting and assembly machinery (SAM), which are responsible for nuclei-encoded proteins import within mitochondria or insertion into the mitochondrial outer membrane (MOM) (Figure 1) [9]. β-Barrel proteins also exist in mammalian mitochondria such as VDAC1 and 2, SAM50, and TOM40 [10]. Both TOMs and SAMs have been implicated in both macromolecules transport and apoptosis [11].

TOM complexes have evolved from a simple machinery (TOM40, the import channel, with or without TOM70, a protein receptor) to highly structured and dynamic macro-complexes with additional TOM receptor proteins (TOM20 or TOM22) and small TOMs (TOM5, TOM6, TOM7) implicated in assembly and disassembly of the supramolecular TOM complex [8]. TOM complexes are usually used by α-helical proteins to reach the different sub-mitochondrial compartments. Insertion of β-barrel proteins is less characterized, especially in mammals, but two proteins—metaxins 1 and 2 (Mtx1 and Mtx2)—are instrumental in the insertion of these proteins in MOM [12]. The importance of the import process has been highlighted by recent works that have shown that TOM dysfunction is associated with numerous pathologies [13]. Due to the importance of protein import in mitochondrial homeostasis, a link between apoptosis and defective mitochondrial import is likely to occur. However, very few works have explored this connection. TOM20 has been implicated in the import of the apoptotic inhibitor survivin, which is imported into mitochondria upon apoptotic stimuli to inhibit caspase activation [14]. It was shown that TOM70 controls Sendai virus (SeV)-induced apoptosis [15] and mediates granzyme B entry into mitochondria, which enhances its pro-apoptotic activity [16]. In addition, the import machinery can be involved indirectly in cell death through the control of mitochondria integrity/quality and thus related to metabolic, epigenetic, unfolded protein response, radical oxygen species dysregulations.

## 3. Apoptosis and Cell Death

Apoptotic cell death plays a major function in development and in tissues homeostasis in animals. It is now well-established that alterations of apoptosis are among the first alterations leading to cancer. Apoptosis is also crucial in the response to anti-tumoral treatments, and defective apoptosis is a major cause of therapy failure ([17,18] for reviews).

Bcl-2 has been identified in the mid-1980s as an oncogene responsible for the increased cell survival in follicular lymphomas because of its anti-apoptotic function [19]. In the following years, a large set of proteins both structurally and functionally related to Bcl-2 have been identified and called “Bcl-2 family” ([20], for review). Although initial experiments tended to associate the function of Bcl-2 to the Endoplasmic Reticulum (ER) [21], it was rapidly observed that the main function of Bcl-2 family members was to regulate the permeability of the mitochondrial outer membrane (MOM) to different proteins, called together “apoptogenic factors” [22]. These factors are released from the mitochondrial intermembrane space to other compartments, including the cytosol and the nucleus, where they exert different functions that give the cell its apoptotic characteristics. These apoptogenic factors include cytochrome c [23], Smac/DIABLO [24], Omi/HtrA2 [25], AIF [22], and endonuclease G [26]. Of note, cells may die from accidental cell death (ACD) or regulated cell death (RCD) [27]. RCD is a process essential for tissue homeostasis during normal and developmental periods. RCD is dedicated to the elimination of old or potentially dangerous cells and can be interrupted or reversed—at least some steps of it—while ACD is definitive and irreversible, triggered by severe stimuli.

For 30 years, the knowledge regarding Bcl-2 family members and how they regulate apoptosis was extensively improved by several teams. It is now established that the pro-apoptotic members of the family—namely, Bax, Bak, and Bok—are able to form and/or activate large pores in the MOM that display a size large enough to enable the release of apoptogenic factors, which was identified by electrophysiology [28,29] and microscopy [30,31]. Anti-apoptotic proteins of the family—namely, Bcl-2, Bcl-xL, Mcl-1, and Bfl-1—prevent the process of formation/activation of the pore, but it is unclear whether they also inhibit it once it is formed [32]. The family also includes a large set of regulators, collectively called “BH3-only proteins” that may activate pro-apoptotic proteins (such as Bid, Bim, or Puma) and/or inhibit anti-apoptotic proteins (such as Bad or Bmf, or again Bid, Bim, and Puma). Site-directed mutagenesis experiments have provided strong evidence that Bcl-2 family members interact with each other through their homology domains (called BH1 to BH4) (Figure 2). These interactions modulate their subcellular localization, their ability to interact within the family or with other partners, and their ability to regulate several processes, including MOM permeabilization but also other functions involved in cell survival and death such as Ca^2+^ homeostasis [33,34] or metabolism [35,36,37,38].

The precise subcellular localization of Bcl-2 has always been a matter of debate. As mentioned earlier, most of Bcl-2 family members have been found at least once in almost all subcellular compartments due to the different models and experimental conditions used. However, since their most established function in apoptosis is the permeabilization of MOM, their mitochondrial localization has been the focus of the most attention.

## 4. Do Bcl-2 Family Proteins Actually Need Mitochondrial Receptors?

This may sound like an unexpected question in a review focused on the role of mitochondrial receptors in the mitochondrial localization of Bcl-2 family members. Indeed, the first studies showing the mitochondrial localization of these proteins did not even consider the possibility that the mitochondrial import machinery could be involved in the mitochondrial localization of Bcl-2 family members, even though this machinery was already extensively described both at the molecular and functional levels. Instead, most investigators considered that Bcl-2 family members contained all the structural information needed to reach mitochondria by themselves [39]. A large number of investigations to identify the intrinsic structural components required for the localization of Bcl-2 family members in the MOM have then been undertaken.

### 4.1. Bcl-2 Family Members vs. Bcl-2 Homologs

After the identification of Bcl-2 as an oncogene [19], the recognition that Bcl-2 was the founding member of a larger set of proteins took several years and was based on the identification of four homology domains called BH1 to BH4. Bcl-xL was identified in 1993 as a close functional and structural homolog to Bcl-2, although a shorter variant, Bcl-xS, was identified as having an opposite pro-apoptotic function [40]. Another anti-apoptotic protein, Mcl-1, was identified the same year, which also displayed striking homologies with Bcl-2 [41]. Still in 1993, Bax was identified as a physical partner to Bcl-2 and characterized as a pro-apoptotic protein [42], similar to Bak, two years later [43]. This led to the “rheostat” model, stating that the balance between pro- and anti-apoptotic proteins of this family regulated the balance between death and life, based on the observation that they could interact with each other through their BH domains [44]. In parallel, the *C. elegans* death regulator Ced-9 was identified as an anti-apoptotic Bcl-2 family member and a strict Bcl-2 homolog [45].

Next, the identification of the first BH3-only protein, Bid, added a level of complexity by introducing a new level of regulation of the interaction between pro- and anti-apoptotic proteins [46]. In spite of the fact that Bid did not have BH1, BH2, and BH4 domains, its general predicted structure [47] was very close to that of Bcl-xL [48], Bax [49], and Bcl-2 [50], establishing that the homology within the family was much wider than the amino acid sequence conservation in the BH domains alone.

A large number of BH3-only proteins have further been identified, which regulate anti-apoptotic proteins (inhibiting their function) or pro-apoptotic proteins (activating their function) or both [51]. However, none of them displayed the same general structure as Bcl-2, Bax, or Bid. This led to the hypothesis that Bcl-2, Bcl-xL, Mcl-1, Bax, Bak, Bid, and several other proteins are actual Bcl-2 homologs having a common ancestor, while most BH3-only proteins regulating apoptosis (Bad, Bim, Puma, Noxa, etc.) could be the result of a convergent evolution [52]. Interestingly, other proteins having a distantly related BH3 domain have been identified, displaying different functions, such as the autophagy regulators Beclin-1 [53] and Nix [54].

### 4.2. The Positive Charges in the C-Terminal End of Bcl-2 and Bcl-xL

In addition to the BH domains, and with the notable exception of Bid, a common feature of the main Bcl-2 homologs is the existence of a potential hydrophobic C-terminal α-helix (Table 1). Obviously, this helix is not conserved in terms of amino acid sequence but is conserved in terms of its potential ability to form a membrane anchor. It was hypothesized that the membrane insertion of this helix was necessary and sufficient to drive the localization of Bcl-2 family members not only in the MOM but also in other membranes such as the ER or the nuclear envelope. This was supported by very solid evidence, such as a study by Kaufmann et al. [55]. Fractionation assays showed that both Bcl-2 and Bcl-xL (both endogenous and ectopically expressed) were mostly localized in heavy membranes (i.e., mitochondria), with additional localization in lighter membranes for Bcl-2. Conversely, the two proteins deprived of their C-terminal helices mostly displayed a cytosolic localization. In addition, the minor fraction of truncated proteins found in membranes was removed by an alkaline treatment, showing that they were not membrane inserted. This study showed with little doubt that the C-terminal α-helix of both Bcl-2 and Bcl-xL was needed for their membrane localization and insertion. Furthermore, this study also showed a difference between Bcl-xL, that is exclusively mitochondrial, and Bcl-2, that is more ubiquitous. This last point raised two intriguing questions: (1) is there any characteristic of their respective C-terminal α-helix that explains the difference between Bcl-2 and Bcl-xL, and (2) is the localization of Bcl-2 in mitochondria vs. other membranes regulated, and if yes, how?

The flanking residues of the C-terminal α-helices of Bcl-2 and Bcl-xL are very different. Specifically, the sequence immediately preceding the helix (that the authors called the “X domain”) contains more charged residues (mostly positive, but also negative) in Bcl-xL than in Bcl-2 [55] (Table 1). In the same study, it has been shown that switching the X-domains between the two proteins changed their membrane selectivity. Furthermore, suppressing two of the four positive charges of the X-domain of Bcl-xL decreased its selectivity for mitochondria. Note that similar selectivity experiments have been done with these C-terminal sequences fused to GFP with similar results, suggesting that the information required was indeed entirely present in this C-terminal domain. A thermodynamic study indicated that the two positive charges flanking the N-side of the α-helix of Bcl-xL might not be crucial for membrane anchoring but might be required for an adequate orientation of the helix within the membrane [56].

Concerning the second question, the authors showed that mutating the two C-terminal Bcl-2 residues from HK to RK, thus increasing the net positive charge at neutral pH from ~1.5 to ~2, increased the mitochondrial selectivity of the protein, suggesting a strong interaction with negatively charged phospholipid heads on the intermembrane side of the MOM. Conversely, removing one of the two positive charges of Bcl-xL significantly decreased the selectivity of Bcl-xL for the MOM [55]. From these data, it appears that the higher density of positive charges both in the “X domain” and at the C-terminal end of Bcl-xL, compared with Bcl-2, is associated with a greater selectivity for the MOM. It follows that the localization of Bcl-2 might be more variable and then more subjected to cellular regulations than the localization of Bcl-xL. Although it had not been investigated in this study, the anti-apoptotic protein Bcl-w also seems to contain a “X-domain” and a single positive charge at its C-terminus. The question remains about whether and how a modulation of the mitochondrial selectivity of Bcl-2 may occur in vivo? This is addressed below.

### 4.3. Bax

In spite of the presence of a similar (but not identical) C-terminal hydrophobic α-helix in Bax, the behavior of the protein is completely different from both Bcl-2 and Bcl-xL. For a long time, Bax has been considered to be exclusively cytosolic in non-apoptotic cells. This was largely based on a striking study by Youle’s group showing the relocation kinetics, from cytosol to mitochondria, of a GFP-Bax fusion protein [57]. This relocation process was further confirmed by a large number of investigations using a wide range of methods and is now considered as an established event in the early steps of apoptosis ([58], for review). The obvious question is why the existence of a hydrophobic C-terminal α-helix in Bax did not lead, similar to Bcl-2 and Bcl-xL, to a constitutive membrane localization. In vitro binding assays showed that, contrary to Bcl-xL, Bax did not bind spontaneously to isolated mitochondria [59]. The replacement of the C-terminal helix of Bax by that of Bcl-xL restored the binding, while the replacement of the C-terminal domain of Bcl-xL by that of Bax prevented the binding. This was in full agreement with the previous proposal that the C-terminal domain of Bcl-xL contained all the required information for mitochondrial binding and that showed that the C-terminal domain of Bax did not contain it. Strikingly, contrary to both Bcl-xL and Bcl-2, Bax does not have any positive charge upstream of the C-terminal α-helix that, according to the hypothesis of the “X domain”, would dramatically impair its selectivity for the mitochondrial membrane.

Bax has a Ser residue, in position 184 that is the target of anti-apoptotic protein kinases, such as AKT [60] or PKCζ [61]. The substitution of Ser184 by a non-phosphorylatable non-charged residue (Ala or Val) induced a constitutive membrane localization of Bax in cells and an increase in in vitro binding to mitochondria. Conversely, the phosphorylation of Bax by AKT, similar to the substitution of Ser184 by a negatively charged residue (Asp or Glu), prevented the mitochondrial localization in cells. It should be noted, however, that the consequences of these changes in Bax localization on its activity were more difficult to interpret because of dramatic changes in the stability of the resulting proteins [62]. It is noteworthy that the deletion of S184 converted the α9 helix of Bax into a strict membrane anchor [49], able to overcome other regulations of Bax addressing [63]. Membrane-inserted BaxΔS184 could, however, serve as a receptor to cytosolic wild-type Bax, showing that its conformation was still compatible with Bax/Bax interactions [63] (see below).

It is noteworthy that Bcl-xL also has a Ser residue at about the same position in its C-terminal α-helix (Ser228, one turn before the end of the helix); however, there is no indication to date that this residue could be the target of a protein kinase.

The function of Bax C-terminal α-helix as a bona fide membrane anchor thus remains an open question. Indeed, the experiments reported above indicated that (wild-type) α9 is not the main driver for Bax membrane insertion, but they did not eliminate it as a major player in the stabilization of active inserted Bax. Structural studies did not help much. Indeed, while NMR studies of soluble Bax suggested that the helix is stabilized in the hydrophobic groove surrounded by the 3 BH domains [49], the X-ray diffraction studies of the Bid BH3-activated Bax dimer was conducted on a protein deprived of the C-terminal helix (for technical reasons linked to purification yield) [64]. Because of the same technical limitations, BaxΔCter has been produced for years and used in reconstitution assays in liposomes with results that were not very different from more recent experiments made with the full-length protein. This suggested that α9 was not a major determinant in the capacity of purified Bax to permeabilize artificial membranes and that other parts of the protein were more directly involved. A most likely candidate was the amphipathic hairpin structure formed by helices α5 and α6 [65], that resembles the structure of bacterial killer toxins [66]. However, experiments aiming at demonstrating this point on intact proteins were not completely convincing since they were based on the analysis of large deletions in the protein [67]. Nevertheless, this hairpin structure has been, for long, the basis of the models of membrane permeabilization induced by Bax, until the X-ray diffraction studies on an activated Bax dimer (without α9) showed that they did not form a hairpin but rather a flat surface stabilizing a head-to-tail conformation of the dimer [64].

It should be noted, however, that the crystal structure of this incomplete dimer might not reflect the dynamics of the protein in cellulo. Additionally, it had been shown before that the forced dimerization of Bax led to an increased insertion that overcame, again, other regulations, suggesting that it may not reflect the physiological situation [68]. Actually, the comparison of the over-expression of full-length Bax vs. BaxΔC in mammalian cells, or their heterologous expression in yeast, showed that BaxΔC was as efficient as full-length Bax to promote the release of cytochrome c [69,70]. However, the sensitivity to Bcl-xL (or Bcl-2) was greatly affected [69,71], suggesting (i) that the release promoted by BaxΔC did occur through a non-selective process, possibly due to the uncontrolled accumulation of the truncated protein in the MOM or (ii) that the C-terminal α-helix of Bax was involved in the interaction with Bcl-xL or Bcl-2. None of these hypotheses can be discarded to date. It is also possible that the loss of efficiency of Bcl-xL on BaxΔC is linked to an abnormal conformation of the truncated protein, limiting the interactions of the BH domains of Bcl-xL and Bax.

The proline residue at position 168 in the short loop between α8 and α9 helices has also been a target of studies. Due to their greater tendency to adopt the cis form, compared with other residues, prolines are more subject to support cis/trans isomerization that may have dramatic consequences on protein conformation. Peptidyl-prolyl cis/trans isomerases catalyze the spontaneously slow (seconds to minutes) conversion from cis to trans within a time range compatible with rapid conformational changes (<100 ns). Considering the critical position of Proline 168, with the potential consequences of its isomerization on the exposure of the hydrophobic α9-helix, extensive mutagenesis studies of this residue have been carried out. Substitution of Proline 168 by alanine (which is almost exclusively under the trans form) has been shown to increase Bax localization in human glioblastoma [71] and yeast cells [72], associated with a greater capacity to release cytochrome c. However, in other studies, it was reported that the same mutant remained in the cytosol [73], where it could form inactive dimers/oligomers [74]. In cell-free assays with isolated yeast or human HCT116 mitochondria or liposomes, the mutant P168A was only marginally more active than the wild type [75]. This suggested that yet unknown cellular factors, absent (or reduced) from in vitro systems and from certain cellular models, might be involved in the conformation of this loop and, consequently, in the movements of α9 helix. Interestingly, it has been reported that a peptidyl-prolyl cis-trans isomerase, Pin1, was able to interact with Bax and modulate its mitochondrial localization [76].

### 4.4. Bak

Cellular studies have established that, unlike Bax, Bak has a constitutive mitochondrial and membrane-inserted localization ([58], for review). The protein is inactive in healthy cells and is activated in apoptotic cells following changes in its interactions with different partners, which inhibit or stimulate the process of oligomerization leading to the formation of a pore having very similar properties to the Bax pore, with dimerization through the amphipathic α6 helix [77]. Similar to Bax, the C-terminal hydrophobic α-helix of Bak is flanked by positive charges on the C-side and a proline residue on the N-side. Unlike Bax, there is not potentially phosphorylatable residue in this helix.

A recent study by HDX-MS showed that the activation of liposomes-bound BakΔC by cBid was associated with an increase in the disorder of the N-terminal part of the protein, leading to a decrease in potential interactions between helix α1 and helices α6–α8 [78]. This suggests that the N-terminal part of Bak has somehow a negative regulatory function on the activation process. Quite interestingly, a similar conclusion had been drawn for Bax on the basis of the existence of an alternative variant, called BaxΨ [79], lacking the 20 first residues, that was spontaneously inserted and active in both glioblastoma cells and yeast [80]. This shows that, unlike anti-apoptotic proteins, the N-terminal part of both Bax and Bak is actively involved in their interactions (in every sense, including insertion, activation, and oligomerization) with the MOM.

### 4.5. Other Bcl-2 Homologs

Among Bcl-2 homologs, two proteins have an obvious C-terminal hydrophobic α-helix. The anti-apoptotic protein Mcl-1 has one positive charge on its C-side, but the residues corresponding to the X-domain of Bcl-2 and Bcl-xL comprise negative charges, which are not expected to favor its insertion in the MOM. However, the localization of Mcl-1 is largely mitochondrial [81], with a role of its C-terminus in this localization [82]. However, similar to Bax and Bak, its N-terminus also seems to be involved in its mitochondrial localization [83].

The protein Bcl2L13 (also called Bcl2-rambo), has been identified as a pro-apoptotic protein [84]. Indeed, its heterologous expression in *Drosophila* promotes apoptosis [85]. However, another study suggested an anti-apoptotic function in adipocytes [86]. Quite unexpectedly, Bcl2L13 has also been identified as a mitophagy effector, distinct from the Pink/Parkin pathway [87]. Strikingly, it has been found that it fully compensated for the absence of the yeast mitophagy regulator ATG32, which is not a Bcl-2 homolog and does not have any BH3-related domain, suggesting that the BH domains of Bcl2L13 have nothing to do with its function in mitophagy. Structurally, the C-terminal helix of Bcl2L13 is more related to Bax and Bak than to Bcl-2 and Bcl-xL, with a X-domain having a global neutral charge.

None of the BH3-only proteins, including Bid, have an obvious C-terminal hydrophobic helix. Puma has a short stretch of 13 hydrophobic residues flanked by positive charges, but in addition to the fact that it seems too short to cross a “normal” membrane (but might cross if it is associated to other proteins), the presence of two proline residues would not help to stabilize an α-helix.

The anti-apoptotic protein Bcl2A1 (or A1/Bfl-1) also falls in this category of proteins that does not have an unambiguous C-terminal hydrophobic α-helix. However, the helix has some amphipathic features that contribute to its mitochondrial localization, albeit with no evidence that it is actually inserted [88].

The pro-apoptotic protein Bok does not have an obvious hydrophobic C-terminal α-helix and is mostly localized in ER and Golgi apparatus [89]. The deletion of the C-terminal sequence prevented the membrane localization, showing that it has a membrane anchor function, in spite of the presence of two positive charges within the helix. It is not known, however, whether it is sufficient to correctly address the protein to these membrane compartments, or whether other domains of the protein are involved.

## 5. Bcl2 Family and Mitochondrial Import Proteins

As discussed above, it seems that C-terminal α-helix region of Bcl-2 proteins might be a membrane anchor, providing this helix is exposed and the presence of positive charges supports the mitochondrial localization. Bcl-xL C-terminus seems to be the most efficient in the addressing to the MOM, whereas those of Bax and Bak might not have a proper orientation in the inactive conformation of the proteins to enable their mitochondrial localization. In the case of Bcl-2 and Mcl-1, the situation might be rated as intermediate, with a weak ability to address the proteins alone.

The first evidence that the mitochondrial receptor TOM20 was involved in Bcl-2 mitochondrial localization was obtained by Motz et al. [90]. In their study, the interaction was dependent on two positively charged lysines in the Bcl-2 C-terminus, which are also present in Bcl-xL, for example. We recently re-examined this process in both mammalian and yeast cells [91]. We observed that Bcl-2 was partly localized in the ER of resting U251 human glioblastoma cells and was relocated to mitochondria-associated membranes (MAM) and mitochondria following an apoptotic stimulus. This process involved a physical interaction with TOM20 but not the C-terminal helix of Bcl-2. The domain of interaction of TOM20 with Bcl-2 was identified as a highly positively charged domain between residues 27 and 44, called TBI (for TOM20 Bcl-2 Interacting sequence). A TBI peptide fused to GFP was able to compete with the Bcl-2/TOM20 interaction. Of note, these observations are specific to Bcl-2 since they were not observed with Bcl-xL.

Chou et al. first showed that Mcl-1 mitochondrial localization was enhanced by TOM70 expression [92]. They identified an internal EELD motif in Mcl-1 that is essential to its interaction with TOM70. Of note, the negative charge of this motif, as well as the positive charge of the TBI, confirms that the interaction between Bcl-2 proteins and TOM proteins is different from that happening between TOMs and nucleus-encoded proteins harboring a positively charged presequence.

Bak is a constitutive mitochondrial protein, yet we demonstrated that this localization is dependent on receptor mitochondrial proteins. In resting glioblastoma cells, Bak was identified as part of three complexes involving proteins from the mitochondrial importation/sorting machinery—namely, VDAC2/Mtx1/Mtx2/Bak, Mtx1/Mtx2/Bak, and Mcl-1/TOM70/Mtx2/Bak [93]. It was shown that non-activated Bak directly interacts with Mtx2 in these complexes, whereas apoptosis induction by TNFα was correlated with the interaction of the activated conformation of Bak with the dephosphorylated form of Mtx1 [93,94]. Strikingly, the down-regulation of Mtx2 inhibited Bak mitochondrial localization in resting glioblastoma cells, while Mtx1 down-regulation inhibited Bak mitochondrial localization under apoptotic conditions [94], and the saturation of isolated mitochondria with anti-Mtx1 antibodies inhibited the MOMP induced by activated Bak [93]. Of note, the dephosphorylation of Mtx1 raised its interaction with Bak while decreasing the Bak/VDAC2 interaction (and inversely) and TNFα-induced Mtx1 dephosphorylation [94]. Bak mitochondrial localization is thus dependent on its interaction with mitochondrial receptors within multiprotein complexes, and the shift of Bak from resting to apoptotic complexes is dependent on Bak conformational change and on Mtx1 dephosphorylation, followed by the insertion of Bak in the MOM and subsequent MOMP. Of note, VDAC2 (which does not belong to the mitochondrial importation machinery) has also been involved in Bax-dependent apoptosis in mice, suggesting an additional regulatory role of this protein in Bax/Bak pore-forming activity; indeed, the deletion of VDAC2 (but not VDAC1) prevented Bax-dependent, but not Bak-dependent, apoptosis [95].

Whereas Bak mitochondrial localization seems to be mostly under the control of proteins from the SAM complex—namely, Mtx1 and 2—Bax mitochondrial addressing rather involves the TOM complex. The role of TOM22 as the mitochondrial receptor for Bax during apoptosis has been a matter of debate. In glioblastoma cells and in isolated mitochondria, TOM22 interacts with Bax Hα1 when activated by tBid, which is necessary for the mitochondrial translocation of the monomeric Bax α from the cytosol to the MOM [63,68]. Bax Hα5–Hα6 then insert into the MOM through Bax transient interaction with TOM40, and Bax can oligomerize into high molecular weight oligomers [68]. In the same work, authors showed that the forced cytosolic dimerization of Bax induces a TOM-independent mitochondrial targeting and insertion into the membrane but results in an incomplete MOMP. Other groups also observed a TOM22-dependent mitochondrial translocation of Bax in *Drosophila* [96] and yeast [97,98], but other investigators observed a TOM-independent translocation in yeast [99] and mammals [100], suggesting that both pathways coexist. The significance of these pathways is discussed below.

## 6. Functional Consequences of Interactions between Import Proteins and Bcl-2 Family

Apoptosis completion is dependent on the MOMP and requires the pore-forming proteins Bax and Bak. Their apoptotic function thus takes place at the mitochondrial outer membrane. The interactions between Bcl-2 proteins and proteins from the TOM and SAM complexes are transient, reversible, and participate in the regulation of Bcl-2 proteins function. It is important to keep in mind that the mitochondrial localization of Bak and Bax is required but not sufficient for the completion of the MOMP. MOMP requires the complete activation of the proteins Bax and Bak, which corresponds to the exposure of epitopes hidden in the non-apoptotic conformations [101,102,103] ([104], for review). Anti-apoptotic proteins exert their function through the binding of Bax and Bak, preventing their oligomerization and/or insertion into the MOM, responsible for the MOMP and subsequent release of apoptogenic proteins into the cytosol. It is now generally recognized that the exposure of Bax and Bak epitopes related to their activated, apoptotic conformation can be transiently detected in healthy cells, in the absence of apoptosis. These reversible modulations in the proteins conformation enable their interaction with other proteins from the Bcl-2 family—namely, anti-apoptotic Bcl-2, Bcl-xL, and Mcl-1. Indeed, the exposure of Bax- and Bak-BH3 domain is required for their binding to anti-apoptotic proteins in the hydrophobic pocket. If there is quite no doubt that the binding of Bax and Bak to mitochondrial receptors should increase the probability of the MOMP to occur, the role of mitochondrial receptors to anti-apoptotic Bcl-2 proteins is less clear. Do they improve their ability to inhibit Bax and Bak or do they promote the mitochondrial co-recruitment of pro-apoptotic members of the family at the site of Bax–Bak pore-forming activity?

In resting glioblastoma cells, Bak interacts with Mtx2 [93]. In one of the non-apoptotic complexes identified, Bak and Mtx2 are found in a complex with Mcl-1 and TOM70. The interaction between Mcl-1 and TOM70 [92] targets Mcl-1 in the neighborhood of proteins from the SAM complex (Mtx1/2). We can thus postulate that the proximity between Mtx2 and TOM70 at the MOM may facilitate the binding of Bak by Mcl-1 and stabilize the non-apoptotic conformation of Bak. It was furthermore established that Bak interaction with VDAC2 following its activation by tBid prevented the formation of high molecular weight oligomers of Bak [105]. In HeLa and GBM cells, VDAC2 and Mtx1 compete for the binding of activated Bak, and the dephosphorylation of Mtx1 induces a shift of Bak binding from VDAC2 to Mtx1 [94]. It thus seems that dephosphorylated Mtx1 is required for the efficient insertion and oligomerization of activated Bak, followed by MOMP, whereas VDAC2 counteracts this oligomerization in a complex devoid of Mcl-1. Mcl-1 also prevents the activation of Bak induced by the direct activation by Bid [106]. Bid-induced activation of Bak would require the mitochondrial targeting of tBid to the mitochondria. This could occur spontaneously or be facilitated through its interaction with a mitochondrial receptor such as MTCH2 [107,108], but no interaction with proteins from the TOM/SAM complexes was described that could sustain the close proximity between Bid and Bak. A role of cardiolipin in Bid mitochondrial targeting was suggested by the use of recombinant proteins in liposomes [109] but was not confirmed in yeast mitochondria, where the amount of cardiolipin is much lower than in reconstituted liposomes [97]. It is more likely that cardiolipin is involved in tBid interaction with mitochondria, rather than Bax [110,111]. Other lipids might be implicated in the interaction of Bax with mitochondria and/or the initiation of MOMP such as sphingolipids [112], prostaglandins [113], sterols [114], or ceramides [115], but this issue is not detailed in this review.

The TOM complex has also been involved in Bim targeting to mitochondria [116]. The direct interaction between Bim and Bak was suggested to play a role in Bak-induced MOMP [117], rather in the formation of Bak oligomers than in Bak activation. The binding of Bim with the TOM complex [116] may thus enable the inhibitory function of Mcl-1 through the binding of Bim or Bak. According to these considerations, Mcl-1 mitochondrial targeting through TOM70 interaction may thus increase its anti-apoptotic function towards Bak-induced MOMP. Of note, despite Mcl-1 stability being weak under apoptotic conditions, the mitochondrial localization significantly increases the lifetime of the protein [91] and thereby enhances its anti-apoptotic ability.

TOM22 interacts with the monomeric Bax α and results in its translocation from the cytosol to the MOM, as described earlier. The subsequent oligomerization and pore formation is TOM40-dependent [68]. By contrast, the forced cytosolic dimerization of Bax induces a TOM-independent mitochondrial targeting and insertion into the membrane but results in an incomplete MOMP. This suggests that Bax homo-oligomerization in cells should rather take place at the MOM by a TOM22/TOM40-dependent pathway. Recent results suggest that discrete Bax structure rather than large super complexes are efficient to release cytochrome c and that this feature is correlated with Bax mitochondrial residence, regulated by both targeting and retrotranslocation [118]. TOM22/TOM40 may thus be this “discrete” insertion pathway of Bax when activated outside the mitochondria, which may include direct activation (by tBid, Bim, or PUMA, for instance) or indirect activation in a non-mitochondrial compartment via a derepressor BH3-only protein (in the cytosol or at the ER, for example). In this context, one could assume that the interaction of anti-apoptotic Bcl-2 proteins with proteins from the TOM complex should raise the binding of Bax and inhibit the pore-forming activity.

Nevertheless, the binding of Bax and Bak by anti-apoptotic proteins from the Bcl2 family creates a consecutive vulnerability called “priming-to-death” since the release of Bak or Bax from anti-apoptotic proteins liberate more active proteins [119], which, in addition, are more localized to the mitochondria [120]. This priming-to-death is well described in cancer cells. This phenomenon is independent of the complex localization, although we can postulate that the liberation of activated proteins will be all the more efficient if the release happens in the proximity of the MOM. In the case of Bak, which is mitochondrial even in non-apoptotic cells, we can speculate that the binding of anti-apoptotic Bcl-2s to TOMs enhances the inhibition of pore formation. The question of priming is quite different for Bax since inactive Bax is mostly cytosolic and needs to translocate to the MOM to exert its apoptotic function and induce MOMP. According to this, the co-recruitment of complexes between Bax anti anti-apoptotic Bcl-2s may sensitize mitochondria to MOMP. This phenomenon of priming contrasts with another interesting regulatory mechanism—namely, Bax and Bak retrotranslocation—first demonstrated by Edlich et al. [121,122]. These authors demonstrate that Bax and Bak localization results from a dynamic shuttling of Bax and Bak between mitochondria and cytosol induced by anti-apoptotic Bcl-2s. According to this mechanism, anti-apoptotic Bcl-2s can thus release Bax and Bak in their inactive conformation. This adds a level of complexity to the question we discuss.

As far as Bcl-2 and Bcl-xL are concerned, both proteins are located at the MOM and at the ER. They share the ability to associate non-specifically to ER or mitochondrial membranes through their C-terminal positively charged residues. This lack of specificity is supported by the similar lipid composition of both membranes. Both are able to increase the amount of mitochondrial Bax [120,123] and inhibit the MOMP. The difference observed resides in the oligomerization status of Bax under non-apoptotic conditions. Indeed, Bcl-2 overexpression induces the formation of Bax oligomers in the absence of cytochrome c release. What could be the basis of this observation? Could this be attributed to Bcl-2 ability to interact with TOM20, contrary to Bcl-xL [91]?

Data obtained in yeast and mammalian cells show that Bcl-xL over-expression increases mitochondrial Bax, whereas the deletion of Bcl-xL C-terminal Hα9 (Bcl-xLΔC) drastically does. Although Bcl-xL is able to retrotranslocate Bax from the mitochondria to the cytosol, Bcl-xLΔC is not [122]. This suggests that Bcl-xL and Bax are co-recruited to the mitochondria and that Bcl-xL then induces their retrotranslocation from the mitochondria [122]. The mitochondrial targeting may involve the recognition of Bax Hα1 by TOM22 when associated with Bcl-xL. Of note, it seems that Bak is also concerned by the Bcl-xL-induced retrotranslocation but to a much lesser extent [121]. Under non-apoptotic conditions, Bcl-xL may promote Bax retrotranslocation to avoid its interaction with TOM40 and pore formation. Under apoptotic conditions, the inhibition of the retrotranslocation would induce Bax oligomerization and MOMP. Contrary to Bcl-xL, Bcl-2 is actively targeted to the MOM by the interaction with TOM20 under apoptotic conditions, prior to cytochrome c release [91], as shown by the mitochondrial translocation of the ER-addressed protein GFP-Bcl2cb5. Bcl-2/TOM20 interaction is not dependent on Bcl-2/Bax interaction since ABT-737 does not counteract Bcl-2/TOM20 interaction under apoptotic conditions [91]. Bcl-2 mitochondrial targeting is therefore not Bax-dependent, unlike Bcl-xL. This does not exclude, however, that Bcl-2-Bax complexes can be recruited to the MOM. The TOM20-dependent targeting of Bcl-2 may support the ability of Bcl-2 to stabilize inactive Bax oligomers in the absence of Hα5–Hα6 insertion. The interaction between TOM20 and TOM22 [124] may stabilize Bax/Bcl-2 interaction and prevent Bax full activation. The stabilization of Bax oligomers, in the absence of complete membrane insertion observed with activated Bax, may be controlled by Bax Hα9 [125]. The disruption of Bcl-2/Bax interaction may be associated with additional modifications (as Ca2+ local concentration increase [126] or lipid intercession [112,113,127]) would thus enable the organization of Bax oligomers into an apoptotic pore. In addition, the experiments conducted in yeast revealed that Bcl-2 is transferred from the MOM to the ER through the MAM under non-apoptotic conditions [91]. It was shown earlier that Bax-induced cytochrome c release was also dependent on the MAM integrity [128]. MAM could thus participate in the regulation of Bcl-2 proteins between ER and mitochondria. The role played by TOM proteins in this context deserves further study. Of note, the somehow paradoxical effect of Bcl-2 mitochondrial over-expression following TOM20–Bcl-2 interaction may also be attributed to a Bax- and Bak-independent effect of Bcl2 on mitochondrial functions ([129], for example).

## 7. Concluding Remarks

The link between mitochondrial import machinery and Bcl-2 family has been found in some systems, but the role of these interactions is still to be firmly established (Figure 3).

As far as anti-apoptotic Bcl-2s are concerned, the question is not so simple. The targeting of these proteins to the MOM may either promote Bax and Bak retrotranslocation and inhibit their oligomerization and pore-forming activity or sensitize the mitochondria to MOMP through the co-recruitment of “primed” pro-apoptotic proteins. Of note, since the mitochondrial localization of Bax and Bak is required but not sufficient to the MOMP, the functional consequences of the interactions between anti-apoptotic Bcl-2s and TOMs are certainly highly dependent on the cellular context, governing the completion of Bax and Bak full activation. The role played by MAM in this regulation is currently under investigation and will certainly improve our understanding of this highly complex regulation network ([134], for an exhaustive review).

A distinction should also certainly be made between the “physiological” apoptosis of normal cells and the one happening in “stressed” or pathologic cells (cancer cells, hypoxic cells, etc.). Whereas anti-apoptotic Bcl-2s should essentially be considered for their role in the regulation of other BH3-containing proteins in the context of major death-inducing stimuli, the interaction of anti-apoptotic Bcl-2s in non-stressed cells may emphasize their non-canonical functions, independent of Bax and Bak, as established for Bcl-2, Mcl-1, and Bcl-xL (see for instance [129,135,136]). These non-canonical functions suggest the provocative hypothesis that the original function of anti-apoptotic Bcl-2s might be independent of Bax and Bak.

## Figures and Tables

**Figure 1 biomolecules-12-00162-f001:**
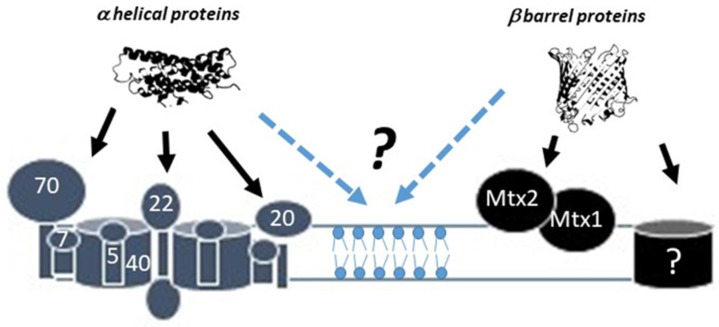
Protein import and insertion in mammalian mitochondrial outer membrane. Overview of the mitochondrial outer membrane (MOM) protein import. Alpha-helix structured proteins are recognized by outer membrane receptors such as TOM70/TOM20/TOM22 either co- or post-translationally. Incorporation into MOM requires a multi-protein structure containing TOM40 and TOM22 plus small TOMs (TOM5, TOM6, and TOM7) (grey structures, left). β-Barrel proteins use a different system called the sorting assembly machinery (SAM) (black blocks, right), which includes 3 proteins in yeast (SAM 35, SAM37, and Sam 50). SAM35 and Sam37 act as peripheral receptors for proteins, while SAM50 is considered as the import/assembly channel. In mammalian mitochondria, metaxin 1 and metaxin 2 (MTX1 and MTX2) are 2 orthologs of SAM 35 and SAM37, and a protein similar to SAM50 has been found. Of note, TOM40 is a substrate of SAM50.

**Figure 2 biomolecules-12-00162-f002:**
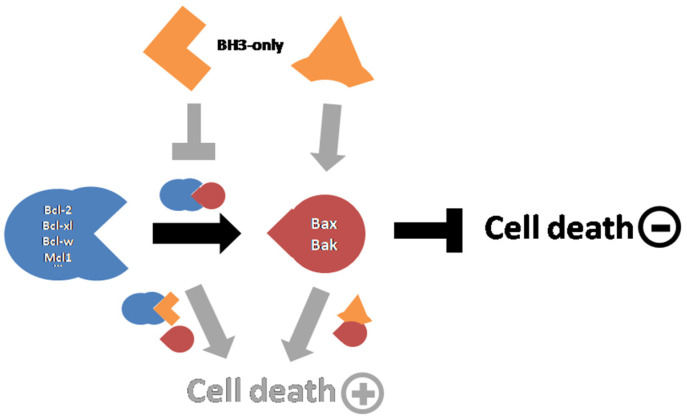
Interplay between Bcl-2 family members. The interaction between antiapoptotic and pro-apoptotic Bcl-2 proteins prevent pore formation and caspase activation responsible for cell death. BH3-only proteins can either counteract the interaction between anti- and pro-apoptotic proteins or directly activate pro-apoptotic proteins.

**Figure 3 biomolecules-12-00162-f003:**
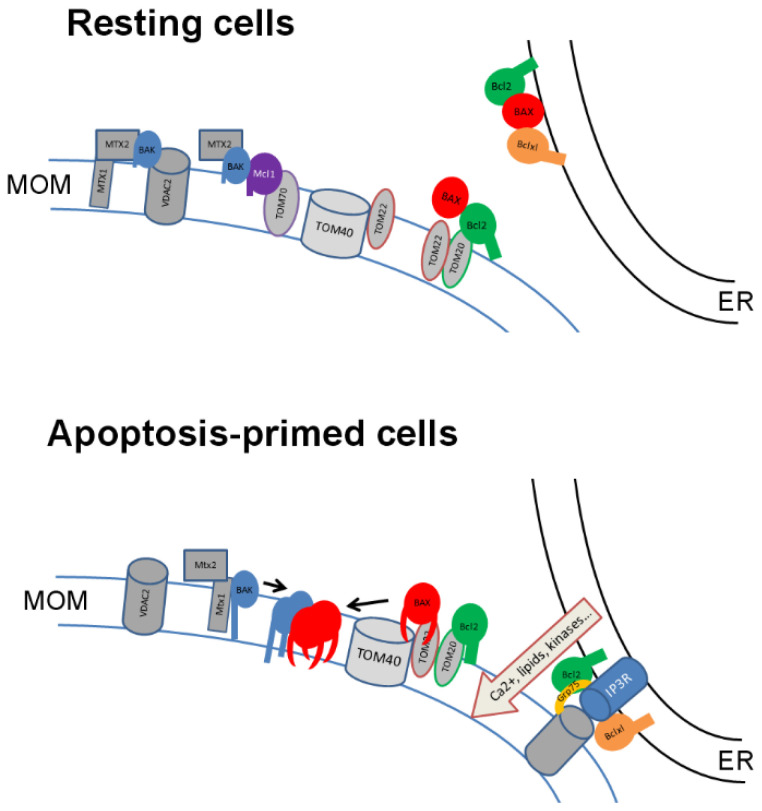
Representation of the interaction network between Bcl-2 family members and the mitochondrial import machinery components during apoptosis commitment. Upper, interaction network in resting cells; bottom, network modifications in apoptosis-primed cells. The ER to mitochondria transfer of lipid, such as PS and ceramides, occurs at ER–mitochondria contact sites [130] and may contribute to the activation of Bax and/or to the formation of the pore. Local concentration of Ca2+/Mg2+ also participates in the regulation of Bcl-2 family interaction with the MOM [126]. The kinase AKT plays a major role both in the reprogramming of mitochondrial metabolism in cancer cells and in the activation of surviving signaling pathways, which includes its capacity to phosphorylate both Bad and Bax ([131], for review). Further, a fraction of active AKT and of its isoforms displays a mitochondrial localization [132,133]. Bax and Bak are the effectors of the MOMP. It is thus quite evident that their interaction with mitochondrial receptors enhances their pro-apoptotic function.

**Table 1 biomolecules-12-00162-t001:** C-terminal sequences of Bcl-2 family members.

Proteins		32 C-Terminal Residues
Proteins having a C-terminal hydrophobic α-helix and an identified “X-domain”	Bcl-2	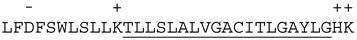
Bcl-xL	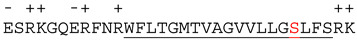
Bcl-w	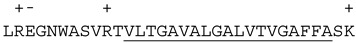
Proteins having a C-terminal hydrophobic α-helix but not yet identified “X-domain”	Bax	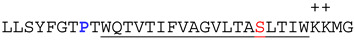
Bak	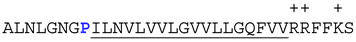
Mcl-1	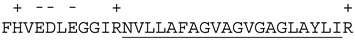
Bcl-2L13(rambo)	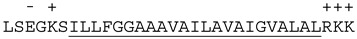
Proteins that do not have a predictedC-terminal hydrophobic α-helix	Bid	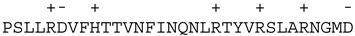
Bim	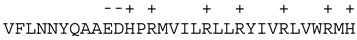
Bad	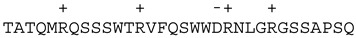
Puma	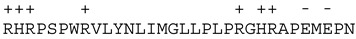
Bcl-2A1(Bfl-1)	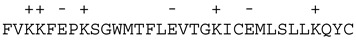
Bok	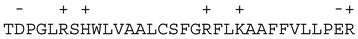

Charged residues are noted with +/− signs. Predicted hydrophobic α-helices are underlined. The proline residues flanking the N-side of the hydrophobic helix of Bax and Bak are indicated in blue. The phosphorylatable Ser residue within the hydrophobic helix of Bax is indicated in red, as is the Ser residue at a similar position in Bcl-xL.

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
