# Peer review of "Bcl-2 Family Members and the Mitochondrial Import Machineries: The Roads to Death"

_biomolecules, 2022, doi:10.3390/biom12020162_

Round 1

Reviewer 1 Report

  1. Despite the figures of the manuscript are described in the test, a more detailed description of the content is required. In the case of Figure 2, I encourage the authors to make the figure more intuitive and visually attractive.

  1. The role of VDAC2 on BAX regulation could be addressed, as it is done for BAK the other canonical apoptotic effector. For example, the work done by Grant Dewson group published in 2018 in Nat. Comm.

  1. Regarding the proapoptotic BID or its active version tBID, the authors stated “Bid-induced activation of Bak would require the mitochondrial targeting of tBid to the mitochondria, although no interaction between tBid and a mitochondrial protein receptor was demonstrated.” I believe this sentence needs to be reformulated or at least to discuss the different studies carried out by Atan Gross lab, where they describe that the mitochondrial carrier MTCH2 is indeed a major facilitator of tBID recruitment to mitochondria. Zaltsman Y., et al, in Nature Cell Biol. 2010, and others.

  1. As for the role of cardiolipin in BCL2 membrane targeting, the authors stated “A role of cardiolipin was suggested by the use of recombinant proteins in liposomes, but was not confirmed in yeast mitochondria where the amount of cardiolipin is much lower than in reconstituted liposomes [96]”. I believe that despite the apparent discrepancy between in vitro models (but also in cellular systems, e.g. Raemy E, et al, 2016 in CDD) and yeast systems, these studies need to be acknowledged.

Of note, not only cardiolipin (or its oxidized isoforms) which is the signature phospholipid of the mitochondria, also hexadecenal, sphingosines, cholesterol and several mitochondrial lipids that have been associated to the membrane targeting of BCL2 proteins, their activation or both. This includes for example: Raemy E, et al 2016 in CDD, Chipuk JE et al 2012 in Cell, Christenson, E et al 2008 in JMB or Lutter M, et al 2000 in Nature Cell Biol. among others.

  1. In relation to BAX/BAK activation in the sentence “Nevertheless, the binding of Bax and Bak by anti-apoptotic proteins from the Bcl2 family creates a consecutive vulnerability called “priming-to-death”, since the release of Bak or Bax from anti-apoptotic proteins liberate fully activated proteins”.

The concept of fully activated is perhaps not very accurate, considering that BAX and BAK are released from the antiapoptotics but not oligomerized nor membrane inserted, which according to the common view corresponds to BAX/BAK activation per se.

  1. As for BAX retrotranslocation and membrane specificity. I believe the sentence “This suggests that Bcl-xL and Bax are co-recruited to the mitochondria and that Bcl-xL then induces their retrotranslocation from the mitochondria [111].” Is misleading, because even it is supported by the cited evidences in the work done by Edlich et al 2011 in cell, BAX and BCLXL are retrotranslocated back together, but that BAX translocation to the MOM (on rate), is not to be influenced by BCLXL.

Author Response

  1. Despite the figures of the manuscript are described in the test, a more detailed description of the content is required. In the case of Figure 2, I encourage the authors to make the figure more intuitive and visually attractive.

We actually omitted to insert explicit legends to figures. We corrected the omission and tried to make figure 2 clearer. 

  1. The role of VDAC2 on BAX regulation could be addressed, as it is done for BAK the other canonical apoptotic effector. For example, the work done by Grant Dewson group published in 2018 in Nat. Comm.

VDAC2 does not belong to the mitochondrial import machinery and our purpose in this review was to focus on the implication of TOM and SAM complexes as mitochondrial receptors from the Bcl-2 family. We mentioned VDAC2 in the context of Bak mitochondrial addressing since it is part from the complexes involving Bak, metaxins and TOM70. We nevertheless added the point that VDAC2 may also be implicated in the regulation of Bax function in the text as required by the reviewer.

  1. Regarding the proapoptotic BID or its active version tBID, the authors stated “Bid-induced activation of Bak would require the mitochondrial targeting of tBid to the mitochondria, although no interaction between tBid and a mitochondrial protein receptor was demonstrated.” I believe this sentence needs to be reformulated or at least to discuss the different studies carried out by Atan Gross lab, where they describe that the mitochondrial carrier MTCH2 is indeed a major facilitator of tBID recruitment to mitochondria. Zaltsman Y., et al, in Nature Cell Biol. 2010, and others.

As discussed above, our purpose was to focus on the TOM and SAM families. Yet the sentence is actually not clear. We corrected the paragraph as required.

  1. As for the role of cardiolipin in BCL2 membrane targeting, the authors stated “A role of cardiolipin was suggested by the use of recombinant proteins in liposomes, but was not confirmed in yeast mitochondria where the amount of cardiolipin is much lower than in reconstituted liposomes [96]”. I believe that despite the apparent discrepancy between in vitro models (but also in cellular systems, e.g. Raemy E, et al, 2016 in CDD) and yeast systems, these studies need to be acknowledged.

Of note, not only cardiolipin (or its oxidized isoforms) which is the signature phospholipid of the mitochondria, also hexadecenal, sphingosines, cholesterol and several mitochondrial lipids that have been associated to the membrane targeting of BCL2 proteins, their activation or both. This includes for example: Raemy E, et al 2016 in CDD, Chipuk JE et al 2012 in Cell, Christenson, E et al 2008 in JMB or Lutter M, et al 2000 in Nature Cell Biol. among others.

The role of lipids in the regulation of Bcl-2 family function is definitely a very interesting issue and we actually published some articles on the subject ourselves. Nevertheless we did not aim at reviewing all the addressing modalities of Bcl-2 family, rather at focusing on the mitochondrial import machinery. We nevertheless added the references to mention this fact.

  1. In relation to BAX/BAK activation in the sentence “Nevertheless, the binding of Bax and Bak by anti-apoptotic proteins from the Bcl2 family creates a consecutive vulnerability called “priming-to-death”, since the release of Bak or Bax from anti-apoptotic proteins liberate fully activated proteins”.

The concept of fully activated is perhaps not very accurate, considering that BAX and BAK are released from the antiapoptotics but not oligomerized nor membrane inserted, which according to the common view corresponds to BAX/BAK activation per se.

 The term indeed is not very accurate. We corrected the sentence as required.

  1. As for BAX retrotranslocation and membrane specificity. I believe the sentence “This suggests that Bcl-xL and Bax are co-recruited to the mitochondria and that Bcl-xL then induces their retrotranslocation from the mitochondria [111].” Is misleading, because even it is supported by the cited evidences in the work done by Edlich et al 2011 in cell, BAX and BCLXL are retrotranslocated back together, but that BAX translocation to the MOM (on rate), is not to be influenced by BCLXL.

This sentence and the previous one both refer to citation 111, and not to the work made by Edlich et al. We duplicated the reference label to be clearer.

Reviewer 2 Report

Bcl-2 proteins are a critical component of apoptotic regulation and encompass a diverse set of functions including apoptotic activation, repression and regulation of other Bcl-2 proteins. One aspect that remains the focus of several studies is their targeting to specific membranes in the cell after their activation. The presented review discusses the interactions of Bcl-2 proteins with mitochondrial import proteins and their possible involvement in Bcl-2 targeting.
The article presents a detailed review of the discussed points; however, the structure of the article makes it confusing and hard to follow. Mainly, the abstract states that the focus of the article is on TOM and SAM complexes and their relation to Bcl-2 proteins, but their relation is not discussed until the last two sections of the review. The question posited at the beginning of section 4 (page 4) also seems out of place as no detailed explanation for TOM-dependent interactions had been made up to this point or in within the section where the question is made. 
The provided sections on BAX targeting motifs is appreciated, however, the review will benefit from more coverage of the role of BH3-only proteins in the targeting of Bcl-2 proteins. 
In the case of Bak vs BAX, a direct discussion of what is known for their interactions with TOM proteins would be helpful due to their shared functions but only the former being an integral membrane protein.  
The article also mentions other forms of targeting regulation such as lipids and divalent cations that could be expanded on to provide additional context to apoptotic changes in the mitochondrial membranes during apoptosis, which would also help new readers follow Figure 3. 
Other comments.
1)    Figures need legends. Particularly for Figure 1 where we have to assume that the left side are TOM proteins, a brief description of the function of these subunits would also be helpful.
2)    The sections are misnumbered after section 4
3)    RMN should read NMR (Page 7)
4)    Other typos: 
a.    Charge in calcium should be superscript
b.    “aminoacid” in the second sentence of section 4.2 (page 5) should be “amino acid”

Author Response

Bcl-2 proteins are a critical component of apoptotic regulation and encompass a diverse set of functions including apoptotic activation, repression and regulation of other Bcl-2 proteins. One aspect that remains the focus of several studies is their targeting to specific membranes in the cell after their activation. The presented review discusses the interactions of Bcl-2 proteins with mitochondrial import proteins and their possible involvement in Bcl-2 targeting.
The article presents a detailed review of the discussed points; however, the structure of the article makes it confusing and hard to follow. Mainly, the abstract states that the focus of the article is on TOM and SAM complexes and their relation to Bcl-2 proteins, but their relation is not discussed until the last two sections of the review. The question posited at the beginning of section 4 (page 4) also seems out of place as no detailed explanation for TOM-dependent interactions had been made up to this point or in within the section where the question is made. 

We wanted to emphasize that it took a long time for investigators to consider the possibillity that the Mitochondrial Import Machinery might regulate the mitochondrial localization of Bcl-2 family members

The provided sections on BAX targeting motifs is appreciated, however, the review will benefit from more coverage of the role of BH3-only proteins in the targeting of Bcl-2 proteins. 

The focus of the review is the role of the mitochondrial import machinery in Bcl-2 family addressing and function. We did not aim at reviewing the structural modifications required for the activation of Bax and Bak and only considered their activation in regard to their interactions with proteins from the TOM and SAM complex. We hopefully clarified this point in several parts of the manuscript.

In the case of Bak vs BAX, a direct discussion of what is known for their interactions with TOM proteins would be helpful due to their shared functions but only the former being an integral membrane protein.  

We tried to clarify this part of the text. We have discussed the role of TOM22 in Bax targeting to the MOM, whereas Bak in involved in several complexes including metaxins and TOM70 depending on its activation status and from metaxin1 phosphorylation.

The article also mentions other forms of targeting regulation such as lipids and divalent cations that could be expanded on to provide additional context to apoptotic changes in the mitochondrial membranes during apoptosis, which would also help new readers follow Figure 3. 

We added a more detailed legend for figure 3 to make this regulation clearer and hopefully answered the reviewer’s comment.

Other comments.
1)    Figures need legends. Particularly for Figure 1 where we have to assume that the left side are TOM proteins, a brief description of the function of these subunits would also be helpful.
2)    The sections are misnumbered after section 4
3)    RMN should read NMR (Page 7)
4)    Other typos: 
a.    Charge in calcium should be superscript
b.    “aminoacid” in the second sentence of section 4.2 (page 5) should be “amino acid”

We corrected the mistakes the reviewer has noticed in our manuscript.

Reviewer 3 Report

Lalier et al. provided a comprehensive review of mitochondrial import mechanisms and their role in regulating the apoptotic signaling by the BCL-2 protein family members. 

Even though the text is scientifically sound, it is very hard for the readers to follow the contexts explained due to grammatic structure of sentences and typos. I suggest extensive editing of English by the authors before acceptance of the manuscript.

Author Response

The review had been edited by an English native speaker. We made some corrections and hopefully made it easier to read.